# RNA-Seq Based Transcriptomic Analysis of Bud Sport Skin Color in Grape Berries

**Wuwu Wen [2,†], Haimeng Fang [1,†], Lingqi Yue [1,†], Muhammad Khalil-Ur-Rehman [3], Yiqi Huang [1], Zhaoxuan Du [1], Guoshun Yang [1] and Yanshuai Xu [1,\*]**

1. College of Horticulture, Hunan Agricultural University, Changsha 410128, China
2. School of Agriculture and Biology, Shanghai Jiao Tong University, Shanghai 200240, China
3. Horticultural Sciences, The Islamia University of Bahawalpur, Bahawalpur 62300, Pakistan
\* Correspondence: yx56@hunau.edu.cn
† These authors contributed equally to this work.

**Abstract:** The most common bud sport trait in grapevines is the change in color of grape berry skin, and the color of grapes is mainly developed by the composition and accumulation of anthocyanins. Many studies have shown that *MYBA* is a key gene regulates the initiation of bud sport color and anthocyanin synthesis in grape peels. In the current study, we used berry skins of 'Italia', 'Benitaka', 'Muscat of Alexandria', 'Flame Muscat', 'Rosario Bianco', 'Rosario Rosso', and 'Red Rosario' at the véraison stage (10 weeks post-flowering and 11 weeks post-flowering) as research materials. The relative expressions of genes related to grape berry bud sport skin color were evaluated utilizing RNA-Seq technology. The results revealed that the expressions of the *VvMYBA1/A2* gene in the three red grape varieties at the véraison stage were higher than in the three white grape varieties. The *VvMYBA1/A2* gene is known to be associated with *UFGT* in the anthocyanin synthesis pathway. According to the results, *VvMYBA1/A2* gene expression could also be associated with the expression of *LDOX*. In addition, a single gene (gene ID: Vitvi19g01871) displayed the highest expressions in all the samples at the véraison stage for the six varieties. The expression of this gene was much higher in the three green varieties compared to the three red ones. GO molecular function annotation identified it as a putative metallothionein-like protein with the ability to regulate the binding of copper ions to zinc ions and the role of maintaining the internal stable state of copper ions at the cellular level. High expression levels of this screened gene may play an important role in bud sport color of grape berry skin at the véraison stage.

**Keywords:** grape; bud sport; RNA-Seq; *MYB*

## 1. Introduction

### 1.1. Bud Sport

Many of the fruits we eat every day are extremely heterozygous in nature [1]. The genomes of fruit trees or vines are highly heterozygous, and in order to adapt to the natural environment, some fruit trees gradually develop inbred incompatibility; as a result, some of the excellent characteristics of fruit trees are lost. Most varieties of fruit trees, such as peach, grape, and citrus, are self-compatible. Most varieties of apple, pear, sweet cherry, and other fruit trees are self-incompatible, while male sterility sometimes occurs in grapes. The VviINP1 gene was identified as related to male sterility in grapes [2]. In order to maintain the excellent properties of fruit during production, asexual propagation (cuttings, strips, and grafting) is used to maintain the exceptional characteristics of fruit [3]. Among cultivation processes, some different mutative traits are observed in similar plants [4], and some mutations are stable to inherit and are called bud sport [5].

Plant bud sport is related to somatic cell mutation that occurs in the cells of the meristem of plant buds, usually expressed on branches, leaves, flowers, and fruits. The

phenotype displayed by the bud sport is significantly different from that of the rest of the plant [6]. In general, bud sport is produced by cell division in the apical meristem of plants, which is triggered by mutations in the stable somatic cells of the first single cell and then fills the cell layer and forms a stable chimera [7,8]. Mutation in this cell gradually fill some or all of the meristem tissue during later stages of growth, and the mutation can be transferred to offspring and can enable mutants to reproduce asexually [9]. Bud sport brings certain types of new traits in the plant itself, while the original qualities of the plant parents are retained, which shape a new mechanism of genetic mutation [10]. Different quantitative genetic studies have located the SDI 119 quantitative trait locus (QTL) on linkage group (LG) 18, explaining up to 70% of phenotypic variance in the 120 seed content parameters. Looking into the potentials of grape varieties for table purposes, mutation-breeding programs have started for other characteristics using chemical and physical mutagens. This is very important for plants because not only the quality of plants can be improved but also more economic value can be generated [11–13].

At present, researchers and growers have selected bud sport varieties that are related to the early ripening, peel color, fruit size, and disease resistance of fruit trees according to different needs [14]. For example, through natural selection, radiation, or colchicine treatment, bud sports varieties related to early fruit ripening and peel color have been found in apples and grapevines [15,16]. Bud sport varieties with enhanced disease resistance have been found in peaches, plums, strawberries, and citrus [17–20], and varieties with enlarged fruit and doubled chromosomes have been found in bananas and kiwifruit [21,22].

*1.2. Fruit Color*

In fruit trees or vines, especially in apples and grapevines, peel color acts as one of the criteria for judging the ripeness of fruit, which is an important indicator and quality parameter of fruits. Numerous examples of fruit berry skin and flesh types of bud sports were reported [23]; the most common type of bud sport changes the color of the flesh or berry skin. The color change in fruit is mainly related to the change in anthocyanin content. Anthocyanins are secondary metabolites of flavonoids. In plants, flavonoids are believed to have a variety of functions, including defense against light coercion. Anthocyanin compounds play an important reproductive role as attractants in plant–animal interactions [24]. Changes in the contents of anthocyanins and synthetic pathways have been fully studied through many plant experiments [25,26].

According to multifaceted verification, some key regulatory genes in the anthocyanin synthesis pathway were analyzed [27]. In the early stages of the flavonoid biosynthesis process, CHS generates chalcone from the 4-coumarinyl-CoA and malonyl-CoA substrates. Chalcone isomerase catalyzes the formation of naringenin, which is the main metabolite of other synthetic branches of this pathway. Downstream of the flavonoid biosynthetic pathway, anthocyanins and leucine are common key substrates for the synthesis of anthocyanins and proanthocyanidins (PAs). Leucoanthocyanidin dioxygenase/anthocyanidin synthase (LDOX/ANS) can convert leucoanthocyanins to anthocyanidins, and anthocyanidins can be further glycosylated by uridine diphosphate (UDP)-glucose to forming flavonoid-O-glycosyltransferase (UFGT). O-methyltransferases (OMTs) catalyze the formation of O-methylated anthocyanins, such as petunidin, peonidin, and malvidin [28,29].

Most of the fruit and skin colors of different fruits, especially grape berries, are associated with the *MYB* gene regulation of anthocyanins [30–32]. The biosynthesis of fruit anthocyanins is controlled by a unique branching of R2R3 *MYB* transcription factors. Normally, the *MYB* gene interacts with the bHLH transcription factor and the WD40 complex protein to regulate the synthesis pathway of anthocyanins [33]. Studies related to grapes and apples have shown that the change in fruit color is due to the insertion of a reverse transcriptional transposon in the promoter region of *MYB* or is a deletion of the *MYB* gene and its upstream alleles that causes the fruit peel or flesh color change. When the *MYB* gene does not show expression or its related sequence alleles are missing, fruit color cannot change to red, blue, or purple [34].

*1.3. Grape Bud Sport*

Grapes (*Vitis vinifera* L.) are one of the most popular fruits in the world and are usually consumed fresh, as well as in the form of several value-added products. The varieties of grape are diverse, including color, fruit size, fruit type, aroma, and other characteristics that show difference in quality. Among them, color is one of the most important quality attributes for consumers. From the beginning, people have used fresh grapes and wine as a source of transmission to spread grapes all over the world. However, with the development of breeding technology, grape breeding started, and many somatic mutations associated with the quality of grapes have been discovered. Many new grape varieties have been developed through bud sport selection.

In the following figure, the color of line under a variety represents grape peel color: green represents green varieties, red represents red varieties, and black represents black and purple varieties.

The white grape 'Italia' could sport into red grapes of the 'Ruby Okuyama' and 'Benitaka' varieties. The red grape 'Okuyama Ruby' and the white grape 'Rosario Bianco' were crossed to produce the red grape 'Rosario Rosso'. The white grape 'Muscat of Alexandria' and the black-purple grape 'Schiava Grossa' were crossed to produce the black-purple grape 'Muscat Hamburg'. The hybridization of 'Bicane' white grapes and 'Muscat Hamburg' black-purple grapes produced the white grape 'Italia' (Figure 1).

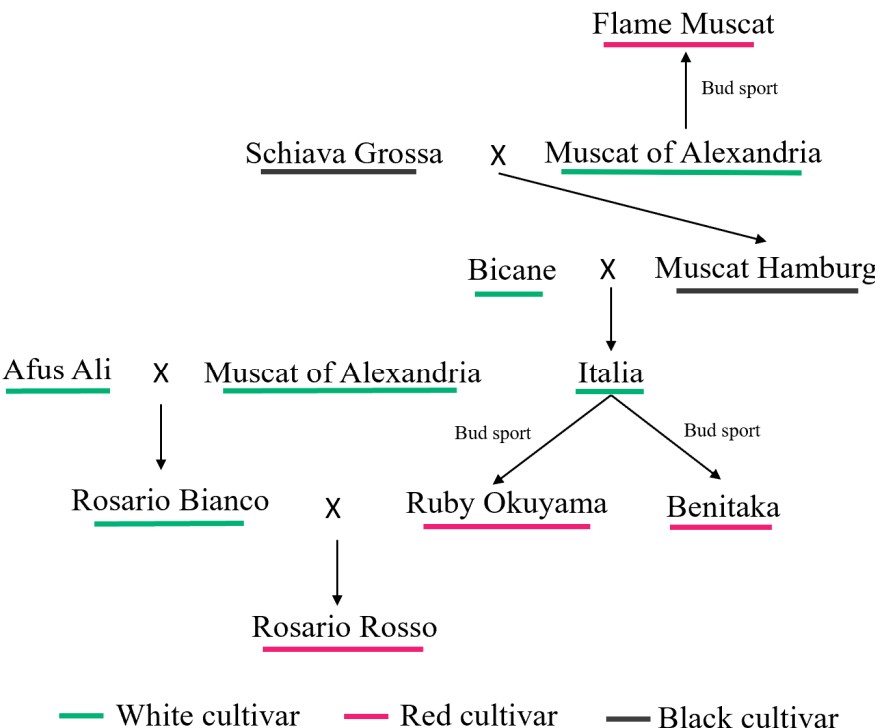

**Figure 1.** 'Italia' is associated with several grape bud sports and related relationship maps.

After thousands of years of natural hybridization and human selection, the color of the berry skins of grapes has become very diverse [35]. According to the presence or absence of anthocyanins in grape berry skin, which is divided into red and black or white varieties, this phenotype is controlled by a single gene locus [29]. There are four *MYBs* at this chromosome with two locations; at least two of these *MYBs* are mutated in white grapes. Either *VvMYBA1* or *VvMYBA2* (or both) can regulate berry peel color. For white grape, two mutations in the coding region of the *VvMYBA2* allele cause its inactivation, while it is not transcribed in white grapes due to the presence of retrotransposons in the promoter region of *VvMYBA1* [36,37]. This results in no accumulation of anthocyanins or very minute accumulation, and the berry skins and flesh color change from dark to

light eventually. However, in some grape bud sport varieties, the deletion of the *Gret1* retransposon restores the function of *VvMYBA1*, and this deletion makes the color of grape berry skins and flesh white to black or purple [38]. However, some studies have shown that, in yellow-green or white bud sports of 'Cabernet Sauvignon' [39], with the exception of *VvMYBA1*, its homologous genes of *VvMYBA2r*, *VlMYBA1-1*, *VlMYBA1-2*, and *VlMYBA2* also regulate the synthesis of anthocyanins. In addition, there are functional and nonfunctional genes among these homologous genes and alleles [26]. Researchers found that, in white grapes, the allele of *VvMYBA1* is homozygous, while the alleles of *VvMYBA1* in red or black grapes are heterozygous [40]. It can be seen in many *MYB*-related genes in berries that play an important role in anthocyanin biosynthesis that the content of anthocyanins and the color of berry flesh and peels might be regulated by these genes.

### 1.4. Transcriptome Sequencing

Bud sport has been studied in many fruits; however, the mechanism of bud sport in grapes remains unclear. In order to understand the mechanism of berry peel color in relation bud sport, we utilize RNA-Seq technology to compare the 'Italia', 'Benitaka', 'Muscat of Alexandria', 'Flame Muscat', 'Rosario Bianco', and 'Rosario Rosso' varieties by selecting samples at 10 wpf (weeks post-flowering) and 11 wpf (12 samples in total). We conclude that, in addition to *UFGT*, the expression of the *LDOX* gene may also correlate with the expression of *VvMYBA1/A2*, and a new gene (gene ID: Vitvi19g01871) that exhibits the highest expression of all the detected genes in white varieties might play an important role at the véraison stage in 'green-red' bud sport berries.

## 2. Materials and Methods

### 2.1. Plant Materials

The research material (berries) used in this study was collected from the vineyard at the Zhengzhou fruit research institute (China) during 2020. The varieties used in the present research were 'Italia', 'Benitaka', 'Muscat of Alexandria', 'Flame Muscat', 'Rosario Bianco', and 'Rosario Rosso'. The vines were 10 years old with 'Y'-shaped tree forms. The berries of each cultivar were in the véraison stage, from 10 wpf (weeks post flowering) to 11 wpf. The red varieties showed notable change in berry color at 11 wpf (Figure 2). Three berries from the upper, middle, and lower parts of each cluster were selected from six uniform clusters. The berry skins were peeled off quickly and frozen in liquid nitrogen immediately. All frozen samples were stored at −80 °C for further analysis.

All the samples were allotted numbers as follows: It10 ('Italia' 10 wpf berries), It11 ('Italia' 11 wpf berries), Be10 ('Benitaka' 10 wpf berries), Be11 ('Benitaka' 11 wpf berries), Ma10 ('Muscat of Alexandria' 10 wpf berries), Ma11 ('Muscat of Alexandria' 11 wpf berries), Fm10 ('Flame Muscat' 10 wpf berries), Fm11 ('Flame Muscat' 11 wpf berries), Rb10 ('Rosario Bianco' 10 wpf berries), Rb11 ('Rosario Bianco' 11 wpf berries), Rb11 ('Rosario Rosso' 10 wpf berries), and Rr11 ('Rosario Rosso' 11 wpf berries) (as shown in Figure 2, respectively).

### 2.2. RNA Extraction and RNA-Seq

RNA was extracted from grape berry skins of different varieties using an RNA extract kit (Solebao Biotechnology Co., Ltd., Shanghai, China). The integrity of sample RNA was detected with agarose gel, the purity and concentration of RNA were detected with a NanoDrop-2000 instrument (Thermo Scientific, Waltham, MA, USA), and the RQN value was tested with Agilent5300 software. Follow-up experiments could be carried out when the RNA was not contaminated by impurities, such as pigment, protein, sugar, etc. The RQN ≥ 7, the brightness of 28/23S was greater than 18/16S, the RNA concentration ≥ 100 ng/uL, the OD260/280 = 1.8~2.2, the OD260/230 ≥ 2, and the total yield of RNA (>1 μg) met the requirements of two RNA libraries.

A Takara RT reagent kit (Takara, Shanghai, China) was used for cDNA and double-strand cDNA synthesis. RNA-Seq libraries were constructed using a TruSeq RNA sample prep kit v2 (Illumina, San Diego, CA, USA). The sequencing process was performed

with an Illumina HiSeq 4000 SBS kit (300 cycles) system (Shanghai Majorbio Bio-pharm Biotechnology Co, Shanghai, China).

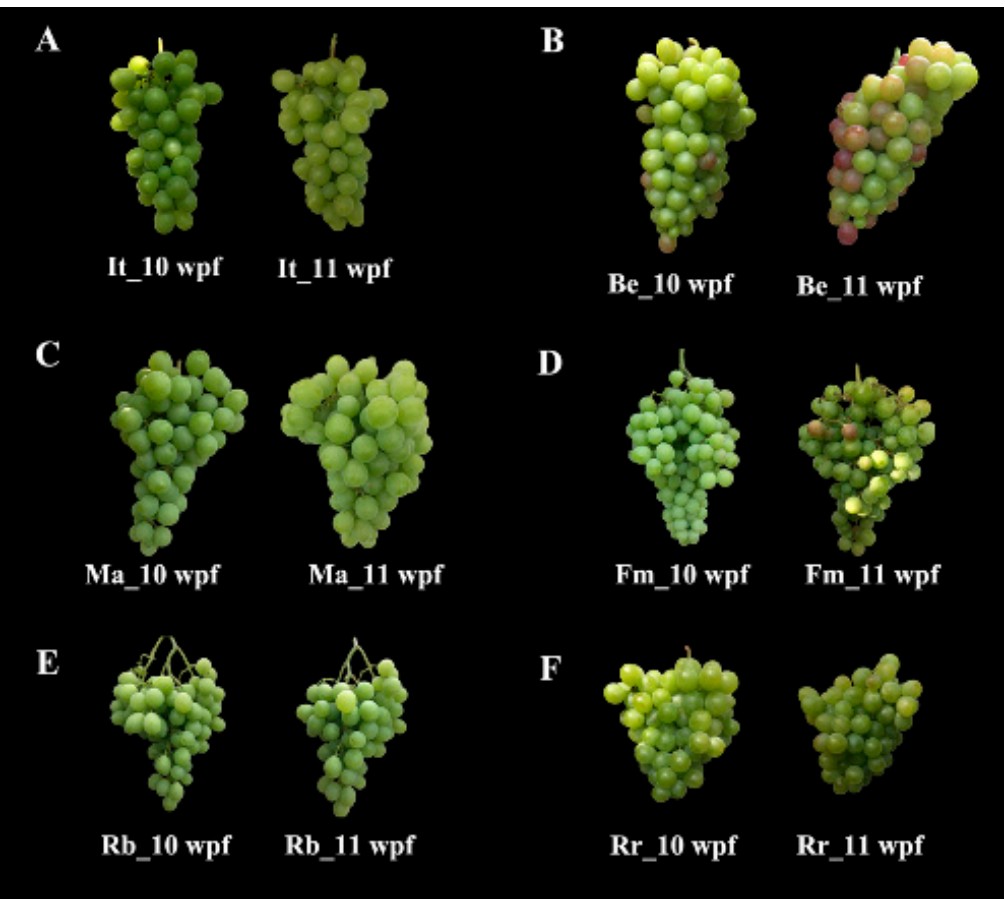

**Figure 2.** Fruit clusters of six varieties during véraison period (10 wpf and 11 wpf). (**A**) 'Italia' berries at 10 and 11 wpf; (**B**) 'Benitaka' berries at 10 and 11 wpf; (**C**) 'Muscat of Alexandria' berries at 10 and 11 wpf; (**D**) 'Flame Muscat' berries at 10 and 11 wpf; (**E**) 'Rosario Bianco' berries at 10 and 11 wpf; (**F**) 'Rosario Rosso' berries at 10 and 11 wpf.

### 2.3. Transcriptome Sequencing and Analysis

SeqPrep (https://github.com/jstjohn/SeqPrep, accessed on 12 February 2022) and Sickle (https://github.com/najoshi/sickle, accessed on 12 February 2022) were used for trimming the adaptors of raw reads and quality control of the raw reads to obtain high-quality reads. The clean reads were aligned to a reference genome (reference genome version: 12X.v2, website source: https://urgi.versailles.inra.fr/Species/Vitis/Data-Sequences/Genome-sequences, accessed on 14 February 2022) with HISAT2 (http://ccb.jhu.edu/software/hisat2/index.shtml, accessed on 14 February 2022) software, and the mapped reads of each sample were assembled with StringTie (https://ccb.jhu.edu/software/stringtie/index.shtml?t=example, accessed on 15 February 2022). To identify DEGs (differential expression genes) between two different samples, the expression level of each transcript was calculated according to the transcripts per million reads (TPM) method. RSEM (http://deweylab.biostat.wisc.edu/rsem/) was used to quantify gene abundances. The DEG analysis was performed using DESeq2/DEGseq/EdgeR with Q values (adjusted *p*-value $\leq$ 0.05, DEGs with $|\log_2 FC| > 1$ and Q value $\leq$ 0.05 (DESeq2 or EdgeR)/Q value $\leq$ 0.001 (DEGseq) that were considered to be significantly different expressed genes. The output of normalized TPM values and the DEG analysis were performed using the Majorbio cloud platform (Shanghai Majorbio Bio-Pharm Technology Co., Ltd.).

### 2.4. Statistical Analysis

A correlation analysis was performed among *VvMYBA1*, *VvMYBA2*, *VvMYB5a*, *VvMYB5b*, *VvMYBPA1*, and grape pericarp anthocyanin synthesis genes using the transcription group TPM value at the level of | r | > 0.7 and *p* < 0.05. Expression level was significantly related to the genes. Pearson's correlation coefficient was used to measure the correlation between two random variables. The closer the Pearson value to 1, the higher the similarity of gene expression between samples, and the better the correlation between the samples.

SPSS v26.0 (Chicago, IL, USA) was used for the significance and correlation analysis of *MYB*-related regulatory genes related to anthocyanin synthesis structural genes data and correlation between anthocyanin synthesis structural genes and *VvMYBA1* and *VvMYBPA1* regulatory genes in two bud sport groups data.

### 3. Results

### 3.1. Quality Control Data Statistics

The total number of raw sequencings reads of each sample ranged from 41,748,704 to 48,476,130 among all the samples. After removing the low-quality reads, the average error rate of the sequencing bases of the clean reads after quality control was less than 0.026%. The percentage of the samples reaching Q20 quality reads was more than 97.74%, and the Q30 percentage was more than 93.32% among all the sequence data. The G and C base ratios were 45.96% and 47.01% of the total bases, respectively. The sequence alignment rates of clean reads matched with the reference genome ranged from 78.27% to 93.11% (Table 1).

**Table 1.** RNA-Seq data quality of all 12 varieties.

| Sample Name | Raw Reads | Clean Reads | Error Rate (%) | Q20 (%) | Q30 (%) | GC Content (%) | Total Mapped |
|:---:|:---:|:---:|:---:|:---:|:---:|:---:|:---:|
| Be10 | 42,647,782 | 42,347,510 | 0.0245 | 98.26 | 94.66 | 46.27 | 34,536,653 (81.56%) |
| Be11 | 45,278,732 | 44,781,572 | 0.0254 | 97.85 | 93.73 | 47.01 | 35,050,465 (78.27%) |
| Fm10 | 45,595,408 | 45,132,012 | 0.0252 | 97.96 | 93.93 | 46.53 | 41,466,312 (91.88%) |
| Fm11 | 42,127,462 | 41,834,580 | 0.0248 | 98.11 | 94.32 | 46.74 | 38,000,195 (90.83%) |
| It10 | 41,804,432 | 41,371,026 | 0.0247 | 98.14 | 94.45 | 46.51 | 38,519,065 (93.11%) |
| It11 | 48,476,130 | 48,148,376 | 0.025 | 98.06 | 94.16 | 45.96 | 44,559,329 (92.55%) |
| Ma10 | 46,362,636 | 46,058,214 | 0.0248 | 98.1 | 94.28 | 46.4 | 41,828,638 (90.82%) |
| Ma11 | 47,352,198 | 46,978,274 | 0.0251 | 97.98 | 94 | 46.36 | 42,739,595 (90.98%) |
| Rb10 | 41,748,704 | 41,378,660 | 0.0252 | 97.95 | 93.95 | 46.35 | 37,569,526 (90.79%) |
| Rb11 | 43,902,394 | 43,576,488 | 0.0249 | 98.06 | 94.18 | 46.04 | 40,136,365 (92.11%) |
| Rr10 | 43,522,364 | 43,177,256 | 0.0249 | 98.06 | 94.22 | 46.48 | 39,821,925 (92.23%) |
| Rr11 | 42,739,842 | 42,458,930 | 0.0246 | 98.19 | 94.51 | 46.57 | 38,767,709 (91.31%) |

(1) Raw reads: the total number of the raw sequencing data; (3) clean reads: the total number of clean sequencing data after quality filtering; (4) error rate (%): the average error rate of the sequencing base corresponding to the quality-filtered data, usually below 0.1%; (5) Q20 (%) and Q30 (%): base or read quality assessment parameters, Q20 and Q30 refer to the percentage of total bases with sequencing qualities of 99% and 99.9% above, respectively. Q20 is usually above 85% and Q30 is above 80%; (6) GC content (%): the percentage of G and C bases corresponding to the quality control data as a percentage of the total bases; (7) total mapped: the number of clean reads that can be matched on the genome.

### 3.2. Differentially Expressed Gene (DEG) Analysis

Through the differential expression analysis of the RNA-Seq data, 3124 DEGs were selected between It11 wpf ('Italia' grape skin samples at 11 weeks post-flowering) and It10 wpf. Compared to Be10 wpf, a total of 2707 DEGs were selected in Be11 wpf. In addition, 1766 DEGs were found between Ma11 wpf and Ma10 wpf, with 1716 DEGs between Fm11 wpf and Fm10 wpf. Rb11 wpf showed a total of 1640 DEG compared with Rb10 wpf, and Rr11 wpf showed 1579 DEGs compared with Rr10 wpf. The number of

upregulated DEGs at the véraison stage (10 wpf to 11 wpf) was greater than the number of downregulated DEGs among the three white varieties of 'Italia', 'Muscat of Alexandria', and 'Rosario Bianco', while in red-colored varieties, 'Benitaka' and 'Flame Muscat' both showed lower upregulated DEG numbers at the véraison stage. Be10 wpf showed 1731 DEGs compared to It10 wpf, Fm10 wpf displayed 2790 DEGs compared to Ma10 wpf, and Rr10 wpf had 2962 DEGs compared to Rb10 wpf. For bud sport varieties in 'Benitaka', 'Italia', 'Flame Muscat', and 'Muscat of Alexandria', more upregulated DEG numbers were found at 10 wpf. Be11 wpf had a total of 2074 DEGs compared to It11 wpf, while 2000 DEGs were screened between Fm11 wpf and Ma11 wpf. Rr11 wpf had 3282 DEGs compared to Rb11 wpf. Among three comparisons of 'Benitaka' versus 'Italia', 'Flame Muscat' versus 'Muscat of Alexandria', and 'Rosario Rosso' versus 'Rosario Bianco', more downregulated DEG numbers were found at 11 wpf (Table 2).

**Table 2.** The numbers of DEGs among difference comparison groups.

| Difference Comparison Group | Total DEG Number | Upregulated DEG Number | Downregulated DEG Number |
|---|---|---|---|
| It10_vs_It11 | 3124 | 1941 | 1183 |
| Be10_vs_Be11 | 2707 | 1114 | 1593 |
| It10_vs_Be10 | 1731 | 1095 | 636 |
| It11_vs_Be11 | 2074 | 925 | 1149 |
| Ma10_vs_Ma11 | 1766 | 1090 | 676 |
| Fm10_vs_Fm11 | 1716 | 505 | 1211 |
| Ma10_vs_Fm10 | 2790 | 1531 | 1259 |
| Ma11_vs_Fm11 | 2000 | 551 | 1449 |
| Rb10_vs_Rb11 | 1640 | 1152 | 488 |
| Rr10_vs_Rr11 | 1579 | 865 | 714 |
| Rb10_vs_Rr10 | 2962 | 931 | 2031 |
| Rb11_vs_Rr11 | 3282 | 936 | 2346 |

### 3.3. Correlation Analysis among Each Sample

The Pearson correlation coefficient between It10 wpf and It11 wpf was close to 1, and It10 showed a positive correlation with It11. The Pearson correlation coefficients between It11 and Ma11 and between It11 and Rb11 wpf were also close to 1. The three white varieties of 'Italia', 'Muscat of Alexandria', and 'Rosario Bianco' showed good correlation (>0.8) at 11 wpf as well. The correlation coefficients between Ma10 wpf and Ma11 wpf and between Rb10 wpf and Rb11 wpf were close to 1, with 'Muscat of Alexandria' and 'Rosario Bianco' closely correlated. The correlation between the three red varieties of 'Benitaka', 'Flame Muscat', and 'Rosario Rosso' was low between 10 wpf to 11 wpf (Figure 3).

### 3.4. Gene Expression Level of VvMYBA1 in Berry Skins

The log2FC value was used to compare the expression levels of *VvMYBA1* in the comparisons of the GC10_vs_RC10 group and the GC11_vs_RC11 group. The results showed that log2FC >7, which means that the expression levels of the *VvMYBA1* gene in the three red varieties were much higher than those in three green varieties. In the comparison of the GC10_vs_GC11 group, the log2FC value was only 1.26, and *VvMYBA1* just reached the differential expression level (if the screening parameter was log2FC > 1.5, then it was not significant). In the comparison of the RC10_vs_RC11 group, the log2FC value was 1.95, and the expression of the *VvMYBA1* gene was significantly different (Figure 4).

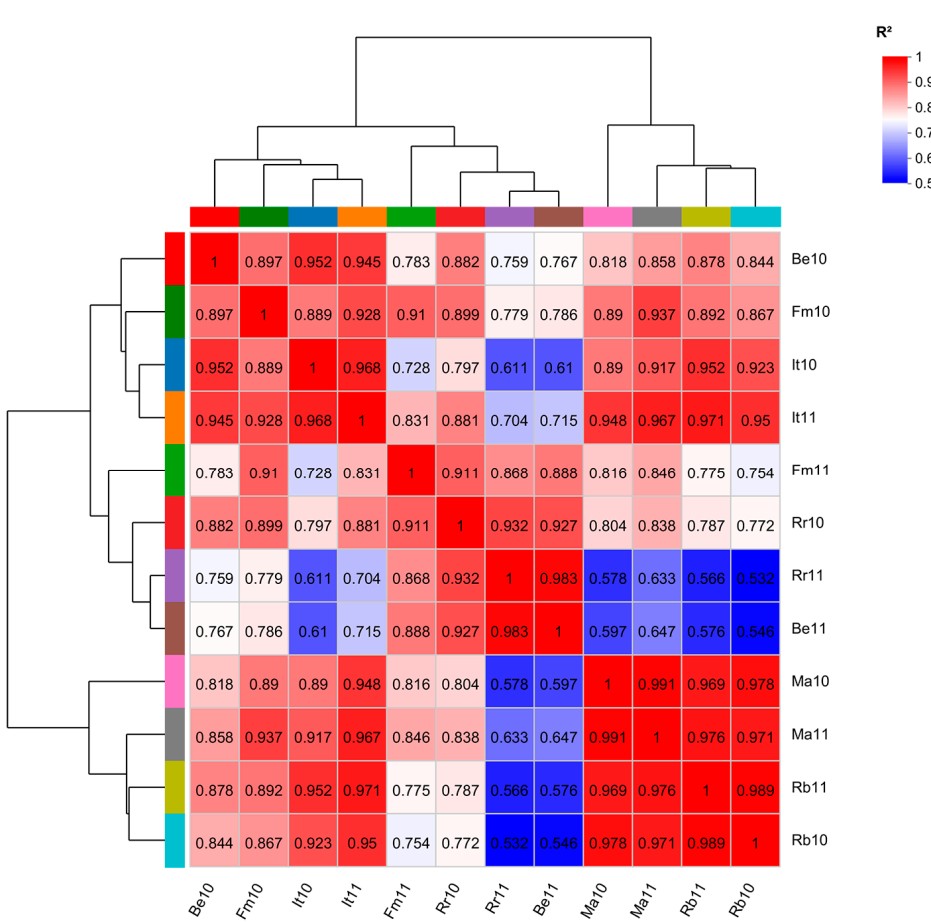

**Figure 3.** Correlation heatmap of six varieties (three groups).

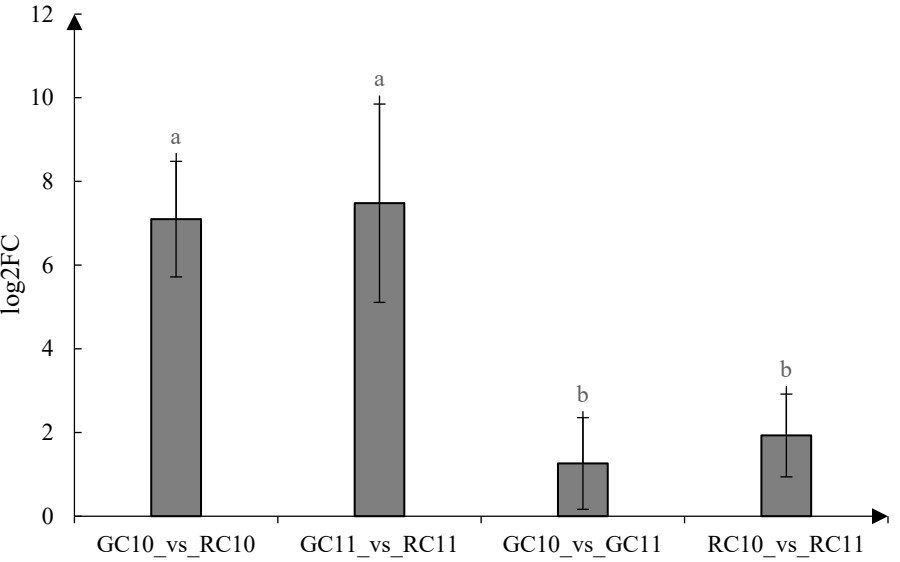

**Figure 4.** Differences in *VvMYBA1* expressions in red and white grape berry skins. 'GC' represents three green cultivars; 'RC' represents three red cultivars. a, b represent the significant level between the data ($p < 0.05$).

*3.5. Anthocyanin-Synthesis-Related Gene Expression Analysis*

Charenone synthase (CHS) is the first key enzyme of the flavonoid pathway. The gene expression in 'Benitaka' was higher at 10 wpf than at 11 wpf, and the gene expression level of Rr was lower at 10 wpf than 11 wpf. The expression level of the CHS-encoding gene

*VvCHS* was significantly different at the véraison stage for 'Benitaka' and 'Rosario Rosso' compared with other varieties. The TPM values of *VvCHS* in berry skins at the véraison stage during the transition period of 'Benitaka' were higher than those of 'Italia' (Figure 5A). Chalcone isomerase (CHI) catalyzed the isomerization of chalcone rings to form colorless flavonoids, and there was no significant difference in the expression of the coding gene *VvCHI* between 10 wpf and 11 wpf for each cultivar (Figure 5B). Flavanone 3-hydroxylase (F3H) is one of the key enzymes in the biosynthetic pathway of anthocyanins, while F3H, F3′H, and F3′,5′H participate in the regulation of two branches of anthocyanin biosynthesis and the F3′H-controlled pathway for the synthesis of red anthocyanins. F3′,5′H, on the other hand, regulates the synthesis of blue-violet delphinidin. The expression level of the F3′H-encoding gene *VvF3′H* was low in each sample, and there was no significant difference between 10 wpf and 11 wpf (Figure 5C). The F3′,5′H-encoding genes of *VvF3′* and *5′H* were not expressed in It 10 wpf and Rr10 wpf, and the expressions of *VvF3′*, *5′H* in the grape berry skins of the two mutated red varieties, 'Benitaka' and 'Flame Muscat', were higher than in 'Italia' and 'Rosario Bianco' (Figure 5D). In addition, the expression level of *VvF3H* in 'Benitaka' was obviously higher than that in 'Italia', and the expression of the F3H-encoded gene *VvF3H* at 10 wpf and 11 wpf for each sample was very low and displayed no difference in each cultivar (Figure 5E). The expression level of the FLS-encoding gene *VvFLS* showed greater variation in the skin of the 'Benitaka' during véraison.

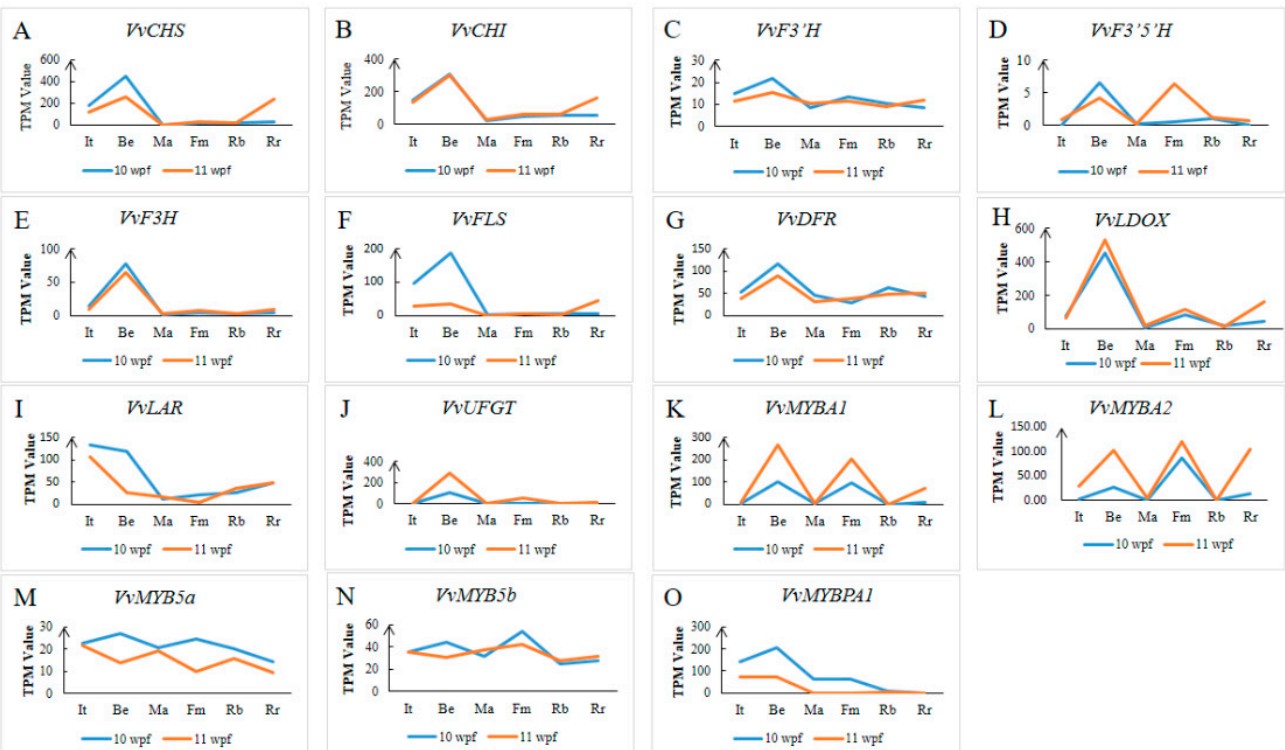

**Figure 5.** Anthocyanin synthesis structure gene and regulatory gene expression analysis of six varieties. (**A**) *CHS*, chalcone synthase; (**B**) *CHI*, chalcone isomerase; (**C**) *F3′H*, flavonoid 30-hydroxylase; (**D**) *F3′5′H*, flavanone3′,5′-hydroxylase; (**E**) *F3H*, flavanone 3-hydroxylase; (**F**) *FLS*, flavonol synthase; (**G**) *DFR*, dihydroflavonol 4-reductase; (**H**) *LDOX*, leucoanthocyanidin dioxygenase; (**I**) *LAR*, leucoanthocyanidin reductase; (**J**) *UFGT*, anthocyanidin 3-*O*-glucosyltransferase; (**K**–**O**) *MYBA1*, *MYBA2*, *MYB5a*, *MYB5b*, *MYBPA1*, transcription factor encode genes, belonging to the R2R3 Myb family, which controls the last steps in the anthocyanins biosynthesis pathway.

Leucoanthocyanidin dioxygenase (*LDOX*) and *UFGT* successively catalyzed the oxidation of colorless proanthocyanidins to form colored delphinidin or anthocyanins and the glycosylation of catalytically unstable anthocyanins to form various stable anthocyanins.

The expression levels of the LDOX-encoding gene *VvLDOX*, the UFGT-encoding gene *VvUFGT*, and the regulatory genes *VvMYBA1* and *VvMYBA2* in the pericarps of the three red varieties were higher than those of the white varieties. Both the *VvUFGT* and *VvMYBA* genes were hardly expressed in the three white varieties during the véraison period (Figure 5H,J–L). The expressions of regulatory genes *VvMYBA5a* and *VvMYBPA1* in the 10 wpf grape berry skins of each cultivar were higher than at 11 wpf (Figure 5M,O).

### 3.6. Correlation Analysis between Anthocyanin-Synthesis-Related Structural Genes and *VvMYBA1* and *VvMYBPA1* Regulatory Genes among Bud Sport Varieties

In the 'Italia' vs. 'Benitaka' bud sport group, *VvMYBA1* and *VvUFGT* showed a significant positive correlation, and *VvMYBA1* may directly regulate *VvUFGT* expression to regulate anthocyanin synthesis. *VvMYBA1* was positively correlated with *VvCHS*, *VvCHI*, *VvF3H*, and *VvLDOX* in the 'Muscat of Alexandria' vs. 'Flame Muscat' bud sport group, while *VvMYBA1* was not significantly correlated with *VvUFGT* (Figure 6A,B). The mechanism of *VvMYBA1* regulation of the anthocyanin synthesis pathway in the pericarp was different between the 'Italia' vs. 'Benitaka' bud sport group and the 'Muscat of Alexandria' vs. 'Flame Muscat' bud sport group. The gene expressions of *VvMYBPA1* and *VvFLS* in the 'Italia' vs. 'Benitaka' bud sport group were positively correlated, and *VvMYBPA1* may directly regulate the expression of *VvFLS*. There was no significant correlation between *VvMYBPA1* and anthocyanin synthesis structural genes in the bud sport group of 'Muscat of Alexandria' vs. 'Flame Muscat', which may not be directly involved in the regulation of anthocyanin synthesis (Figure 6C,D).

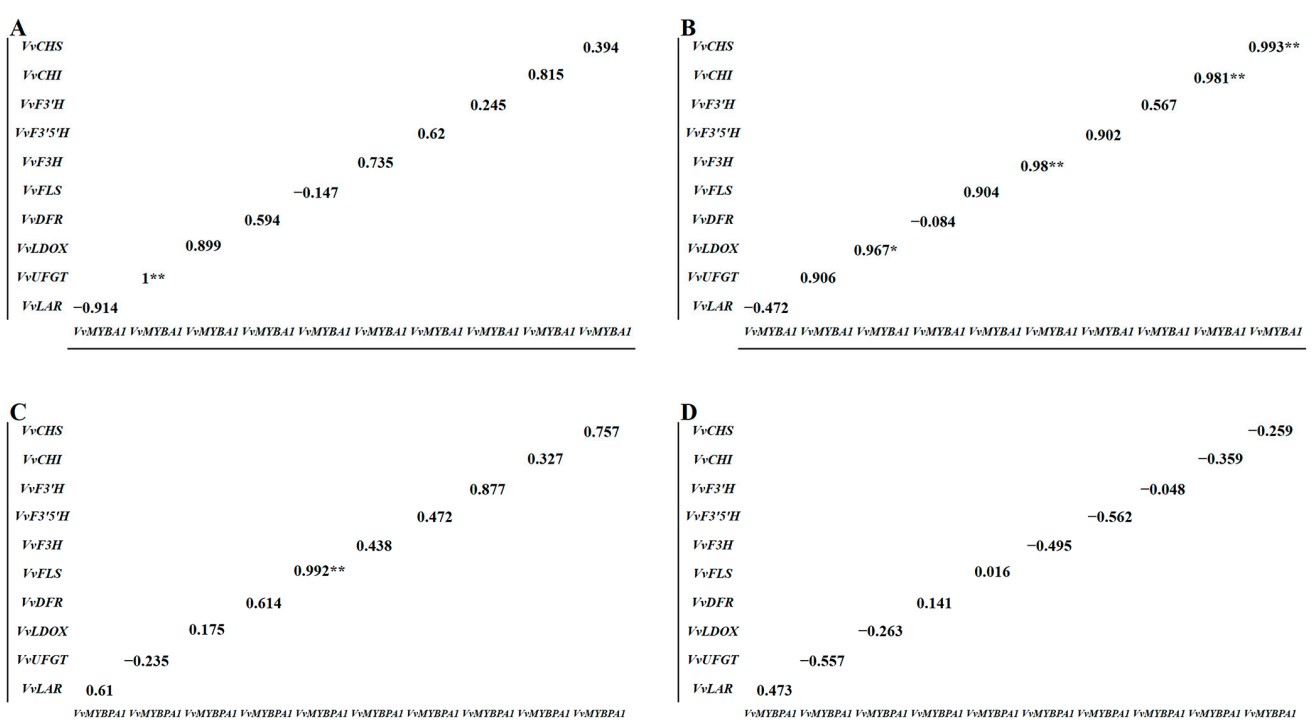

**Figure 6.** Correlation between anthocyanin synthesis structural genes and *VvMYBA1* and *VvMYBPA1* regulatory genes in two bud sport groups. (**A**) Correlation between anthocyanin synthesis structural genes and *VvMYBA1* regulatory gene in 'Italia' vs. 'Benitaka' bud sport group; (**B**) 'Muscat of Alexandria' vs. 'Flame Muscat' bud sport group anthocyanin synthesis structural genes and *VvMYBA1* regulatory gene correlation; (**C**) correlation between anthocyanin synthesis structural genes and *VvMYBPA1* regulatory gene in 'Italia' vs. 'Benitaka' bud sport group; (**D**) 'Muscat of Alexandria' vs. 'Flame Muscat' bud sport group anthocyanin synthesis structural genes associated with *VvMYBPA1* regulatory gene. '*,**' represents the two groups of data reached a significant level, '*' represents $p < 0.05$, '**' represents $p < 0.01$.

### 3.7. Correlation Analysis of MYB-Related Regulatory Genes and Anthocyanin Synthesis Structure

*VvMYB5a* and *VvMYB5b* were not significantly associated with structural genes in the anthocyanin synthesis pathway, which may not be directly involved in regulating the synthesis of anthocyanins. *VvMYBPA1* showed significant correlations with *VvCHS*, *VvF3′H*, *VvF3H*, *VvFLS*, and *VvLAR*, which may directly regulate the flavonoid pathway, anthocyanin synthesis, flavonol synthesis, and catechol synthesis in the anthocyanin synthesis pathway. *VvMYBA1* was positively correlated with *VvF3′5′H*, *VvLDOX*, and *VvUFGT*, which may be directly involved in regulating the synthesis of anthocyanins and regulating the *UFGT* catalytic formation of stable anthocyanin pathways. *VvMYBA2* was not significantly associated with structural genes in the anthocyanin synthesis pathway (Table 3, Figure 7). Among these, the regulation of *VvMYBA2* and *VvMYB5a* was not clear, while synthetic genes regulated by *VvMYBPA1* and *VvMYBA1* were clearly known.

**Table 3.** MYB-related regulatory genes related to anthocyanin synthesis structural genes. ** represents *p* < 0.01.

| Gene | *VvMYB5a* | *VvMYB5b* | *VvMYBPA1* | *VvMYBA1* | *VvMYBA2* |
|------|-----------|-----------|------------|-----------|-----------|
| *VvCHS* | 0.194 | 0.154 | 0.731 ** | 0.384 | 0.231 |
| *VvCHI* | −0.109 | 0.075 | 0.665 | 0.544 | 0.306 |
| *VvF3'H* | 0.459 | 0.487 | 0.839 ** | 0.431 | 0.22 |
| *VvF3'5'H* | −0.09 | 0.33 | 0.349 | 0.747 ** | 0.481 |
| *VvF3H* | 0.24 | 0.184 | 0.709 ** | 0.585 | 0.222 |
| *VvFLS* | 0.468 | 0.263 | 0.873 ** | 0.121 | −0.059 |
| *VvDFR* | 0.231 | -0.09 | 0.663 | 0.374 | 0.007 |
| *VvLDOX* | 0.029 | 0.187 | 0.548 | 0.756 ** | 0.459 |
| *VvUFGT* | −0.152 | 0.007 | 0.304 | 0.831 ** | 0.468 |
| *VvLAR* | 0.5 | 0.078 | 0.752 ** | −0.246 | −0.295 |

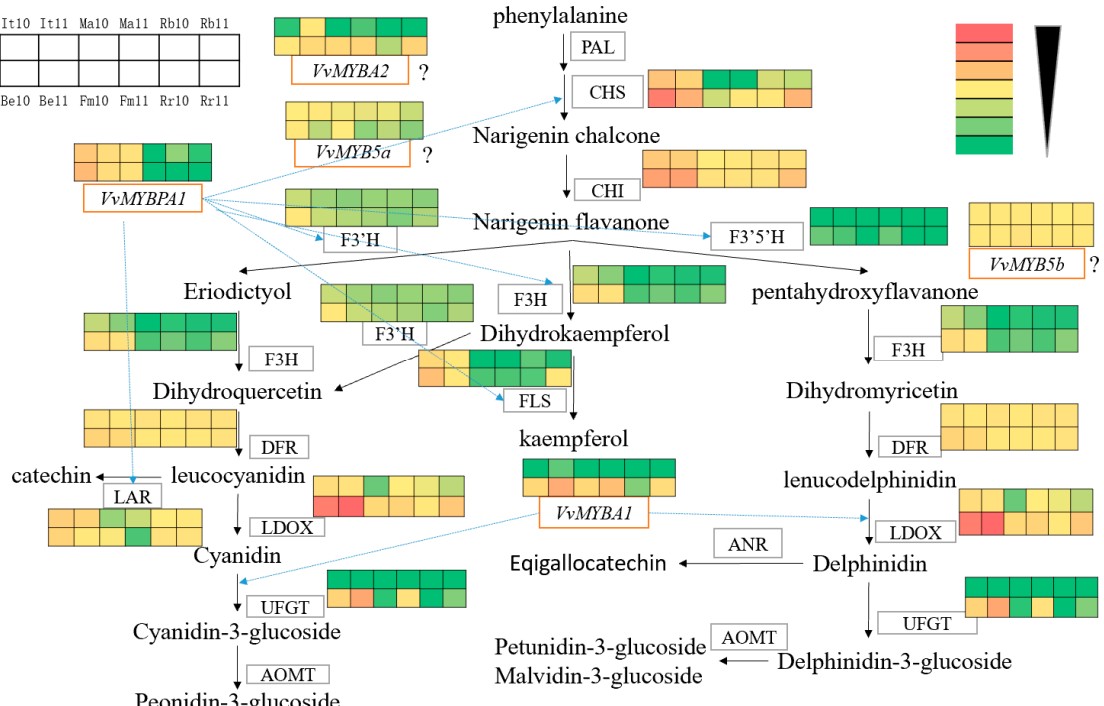

**Figure 7.** The relationships between the expression color scale of anthocyanin synthesis structural genes and regulatory genes in grape peels and the regulation of the *MYBA* gene. The blue dotted arrows represent the structural genes in the anthocyanin synthesis pathway regulated by the regulatory gene; the '?' indicates that the regulatory mechanism of the regulatory gene is not yet clear; the green-to-red color scale means the TPM values showed an increasing trend.

### 3.8. Screening of Genes Involved in the Regulation of Metal Ion Binding

When screening by sorting all gene expression levels (TPM values) in the three white grape varieties, a gene located on chromosome 19 (gene ID: Vitvi19g01871) was found to show the highest expression level. Its expression level was much higher than those of other genes, and the gene was highly expressed (almost the maximum) in the three red varieties. Interestingly, the expression levels of this gene in the pericarps of the three white the three red varieties during the same period were also different (Figure 8). From the comparison of 10 wpf and 11 wpf, this gene was upregulated at 11 wpf compared with 10 wpf in the white cultivar 'Italia', while the opposite was found in 'Muscat of Alexandria' and 'Rosario Bianco' (Figure 8). In the red varieties of 'Benitaka' and 'Rosario Rosso', the expression levels at 11 wpf were downregulated compared with 10 wpf, while the expression in 'Flame Muscat' was higher (Figure 8). According to the functional annotation, it was inferred that this gene encodes a metallothionein-like protein, which regulates the binding of copper ions and zinc ions. Copper ions are related to the synthesis of chlorophyll, which may have a certain impact on the change in peel color at the véraison stage.

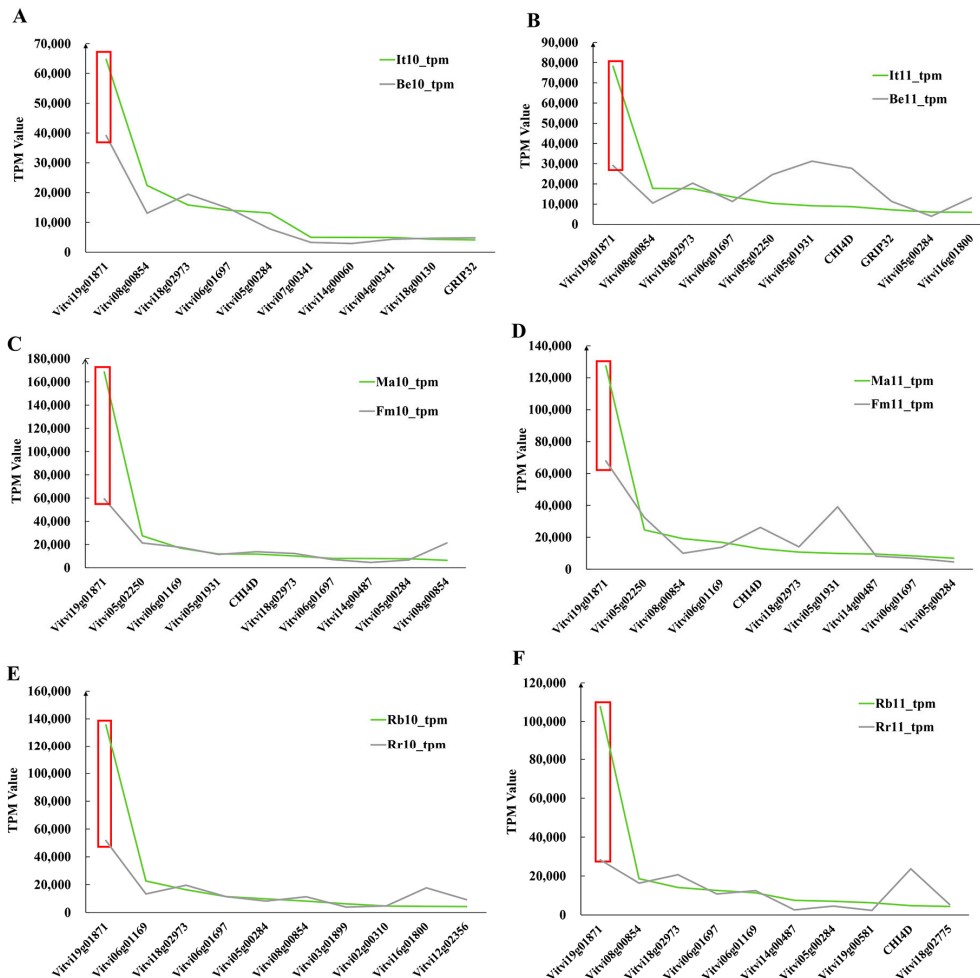

**Figure 8.** (**A**) 'Italia' 'Benitaka' TPM value almost top 10 of gene expression 10 wpf; (**B**) 'Italia' 'Benitaka' TPM value almost top 10 of gene expression 11 wpf; (**C**) 'Muscat of Alexandria' 'Flame Muscat' TPM value almost top 10 of gene expression 10 wpf; (**D**) 'Muscat of Alexandria' 'Flame Muscat' TPM value almost top 10 of gene expression 11 wpf; (**E**) 'Rosario Bianco' 'Rosario Rosso' TPM value almost top 10 of gene expression 10 wpf; (**F**) 'Rosario Bianco' 'Rosario Rosso' Top TPM value almost top 10 of gene expression 11 wpf. The green line represents the TPM values of white cultivars; the black line represents the TPM values of red and black cultivar.

## 4. Discussion

In recent years, it has been observed that many important fruit varieties are selected by bud sport [41,42]. According to statistics, there have been thousands of bud sport types on fruit trees, and some fruit trees can form a variety of bud sports. Due to the particularity of each bud sport, it brings certain characteristics in germplasm resources during fruit production and breeding. Therefore, this is an important approach used in the breeding of fruit crops.

The present study revealed that *VvMYBA1* showed elevated expression levels in the three red sport varieties at 10 wpf compared to three white varieties. In addition, after 11 wpf there were significantly higher *VvMYBA1* gene expression levels compared with the white cultivar grapes (Figure 4B). The *VvMYBA1* gene was proved to be a key transcription factor regulating color change in grape berry skins [43]. The *VvMYBA1* gene was expressed only in red berries, while it was hardly expressed in white berries (Figure 5K).

According to a correlation analysis, the majority of genes or enzymes related to the anthocyanin synthesis pathway were significantly correlated with *VvMYBPA1* and *VvMYBA1* (Table 3). Among them, five genes showed significant correlations with various genes, such as *VvMYBPA1*, *VvCHS*, *VvF3′H*, *VvF3H*, *VvFLS*, and *VvLAR*. Significant correlations of *VvF3′5′H*, *VvLDOX*, and *VvUFGT* with *VvMYBA1* were observed in our study.

The expression of the flavonoid 3-*O*-glucosyltransferase (UFGT) gene is essential for anthocyanin biosynthesis in grapes [44]. The *VvMYBA1* gene normally regulates the expression of *VvUFGT*, a key upstream gene of anthocyanin synthesis [45] considered to be the last step for catalyzing anthocyanin synthesis in the anthocyanin biosynthesis pathway [46], and both are very important in the formation of grape skin color. The RNA-Seq results indicated that the expression trends of *VvUFGT* in the three red varieties were consistent; among them, the expression level in 'Benitaka' was significantly higher than in the other two varieties and was not expressed in white grape varieties (Figure 5J). The above results are consistent with the results of a previous study conducted on 'Italia', 'Benitaka' and 'Flame Muscat' [5]. The Pearson's correlation analysis showed that *VvMYBA1* and *VvUFGT* were highly correlated with the same expression trend (Figure 5J,K). The results also indicated that *VvMYBA1* positively regulated the *VvUFGT* gene and played an important role in the biosynthesis of anthocyanins.

According to previous reports on anthocyanin synthesis in apples and bilberries, it was found that *MYBPA1* could also regulate the expression of *UFGT* [47]. In this experiment, the correlation between these two genes was not high. This may explain why, among the three groups of varieties (the 'Italia' vs. 'Benitaka' group, the 'Muscat of Alexandria' vs. 'Flame Muscat' group, and the 'Rosario Bianco' vs. 'Rosario Rosso' group), *MYBPA1* was not a key transcription factor regulating *UFGT* and the anthocyanin biosynthesis pathway. Its specific regulatory mechanism still needs further study.

*MYBPA1* plays an important role in the anthocyanin biosynthesis pathway, and the expression of *MYBPA1* is positively correlated with anthocyanin accumulation [48]. In blue bilberries, the *MYBPA1* and *MYBA* transcription factors can activate the expression of DFR and ANS genes in the anthocyanin biosynthesis pathway, which are considered key genes for anthocyanin biosynthesis [49]. In this study, the expression levels of the *VvMYBPA1* gene in the two groups of bud sport varieties of 'Italia' vs. 'Benitaka' and 'Muscat of Alexandria' vs. 'Flame Muscat' were higher at 10 wpf compared with 11 wpf, while this gene was not expressed in the 'Rosario Bianco' vs. 'Rosario Rosso' group (Figure 5O). The expression trends of five structural genes (*VvCHS*, *VvF3'H*, *VvF3H*, *VvFLS*, and *VvLAR*, Table 3) related to *VvMYBPA1* were different in the three groups of tested varieties (Figure 5A–E,I). The phylogenetic analysis depicted that the flavonoid-related R2R3 MYBs of *VmMYBPA1* and *VvMYBPA1* belonged to the same group. *VmMYBPA1* could regulate the expression of *CHS* and significantly regulated the expression of the *F3′5′H* gene, while *VmMYBPA1* expression was significantly decreased in white mutant berries compared with blueberries [50], which indicates that it was related to anthocyanin biosynthesis. The expression level of *MYBPA1* is associated with the accumulation of proanthocyanidins (PA) during the early development

of grape berries. The expression level of *MYBPA1* was lower before the véraison stage and peaked at two weeks following the véraison stage, later showing a low expression level. *MYBPA1* activates the promoters of *LAR* and *ANR* in grapes [51]. The expression of *VvMYBPA1* was opposite to that of *VmMYBPA1*, as expressed in 'Italia'. This is in contrast to previous studies showing no expression observed in white grape varieties. A previous study also found that *VvMYBPA1* could also be expressed in seeds [52]. The above results might indicate that the pathway or regulation mechanism of the *MYB* gene in anthocyanin synthesis is different in diverse species.

The gene expression analysis showed that the expression of *VvLDOX* was consistent with the expression trends of *VvMYBA1* and *VvUFGT* in other test materials, except for in 'Italia' (Figure 5H,J–K). *LDOX* has a unique expression pattern in the biosynthesis of anthocyanin in grape peels, and its expression levels were very high in red or black peels, which was related to the content of anthocyanin. *UFGT* is present in many tissues of grape, as well as in the skins of white and red grape varieties, while the expression of *LDOX* is not as absolute as *UFGT* [53]. *VvMYBPA1* was found to activate *VvLDOX* expression in grapes [49], and this result suggested that the expression of *VvLDOX* in 'Italia' may be related to the regulation of *VvMYBPA1*, while there was no significant correlation between *VvMYBPA1* and *VvLDOX*.

In addition to the above results, an interesting point found in this study was the gene located on chromosome number 19 (Gene ID: Vitvi19g01871). The gene expression levels (TPM values) of green varieties at 10 weeks and 11 weeks post-flowering were between 64,711–168,489 and 78,173–127,381, respectively. The expression levels of red grape varieties at 10 weeks and 11 weeks post-flowering were between 39,130–59,249 and 28,319–67,849, respectively. The expression levels of this gene in green varieties were much higher than those in red varieties, as well as much higher than all the other differentially expressed genes (Figure 8). The gene was annotated by GO molecular gene function as a metallothionein-like protein that regulates the binding of copper ions to zinc ions. Copper ions play an important role in the redox of plant respiration and are related to chlorophyll synthesis, which is important for photosynthesis. Increased photosynthesis and chlorophyll lead to excessive chlorophyll accumulation in grape peel cells. However, its mechanism of action in the process of bud sport peel color and anthocyanin synthesis is still unclear, and the function of this gene needs to be further verified at molecular or cellular levels.

## 5. Conclusions

In this study, it was found that *MYBA1/2* and *MYBPA1*, the key genes involved in anthocyanin synthesis in grapes, were highly expressed in red grape varieties, and their expression levels in white grapes were significantly lower than in red grapes. The expressions of *UFGT* and *LDOX* genes were positively correlated with the key peel-color-related gene of MYBA. A newly discovered gene (gene ID: Vitvi19g01871) in this study may play a key regulatory role in grape skin coloration.

**Author Contributions:** Experimental design, Y.X. and W.W.; data curation, H.F. and L.Y.; figures and table making, Y.H. and Z.D.; writing, H.F. and W.W.; manuscript review and editing, M.K.-U.-R. and G.Y. All authors have read and agreed to the published version of the manuscript.

**Funding:** This work was supported by grants from the National Natural Science Foundation of China Youth Fund (No. 32002015), the Natural Science Foundation Youth Fund of Hunan Province (2021JJ40252), and the Hunan Provincial Department of Education Scientific Research Project—General Project (19C0926).

**Data Availability Statement:** Not applicable.

**Conflicts of Interest:** The authors declare no conflict of interest.

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
