# Peer review of "RNA-Seq Based Transcriptomic Analysis of Bud Sport Skin Color in Grape Berries"

_horticulturae, doi:10.3390/horticulturae9020260_

Round 1

Reviewer 1 Report

Change in text the words:

Ve’raison: véraison

Tree: vine à As a general rule, “trees” are woody plants 13 feet tall or taller that typically have only one trunk. A “vine” is a plant whose stems require support. It either climbs up a tree or other structure, or it sprawls over the ground. Vines can climb with tendrils or with other “grasping” appendages, or by coiling their stems.

Description of the differents figures: description and Note must be merge, they can be divide with a period. I recommend to delete the word "Note", is easy to interpret that is a description.

Introduction

1.2. Fruit Color.

The grapes are not trees, are vines!

77-78: Problems in the redaction of O-methyltransferases

1.3 The grape bud sport

98: comma mistake

1.4. Transcriptome sequencing 131

132-139: Unnecessary text.

General comment: all data is very interesting, but it could be merge in a unique text.

2. Materials and Methods

2.1. Plant materials

Delete: (their relationship was mentioned in Figure 1)

2.2. RNA extraction and RNA-Seq

How do you measure the Integrity of Your RNA Samples? The Nanodrop give us a idea of quantity and purity of nucleic acids, but not the quality. Did you use a Bioanalyzer or a Agarose Gel?

2.3. Transcriptome sequencing and analysis

What about the setting the Fold-change and p-val? I consider better to use a FDR 0.05 and a Log2FC of 1.5x to filter the non-differential express genes.

2.4. Pearson correlation analysis AND 2.5. Statistical analysis:

Should be merge if you use the same package.

3. Results

3.1. Quality control data statistics

Table 1: may include a column of the accession number.

3.2. Differentially expressed genes (DEGs) analysis

244: innecesary explanation in Note “the ‘vs’ represents versus.”

3.4. The gene expression level of VvMYBA1 in berry skins

You should use Log2FC for compare expression between samples. Explanation: TPM (Transcripts Per Million) refers to how much RNA is present in a sample. For example, a Log2 TPM of 9 means that for every million transcripts in your sample, 2^9 of them are from gene A. It is the expression level of gene A in a sample.Log2Foldchange describes how one sample is different from another. In this case it is saying that the expression level in Condition 2 is 8 (2^3) times as high as it is in Condition 1.

Please introduce the number of accession for the study to see the raw data sequence

Author Response

Dear editors or reviewers:

We have modified the parts that can be modified. Thank you for your patience. Thank you!

Yanshuai Xu

Reviewer 2 Report

Review report for

„RNA-seq based transcriptomic analysis of bud sport skin color in grape berries”

by Wuwu Wen, Haimeng Fang, Lingqi Yue, Muhammad Khalil-Ur-Rehman, Yiqi Huang, Zhaoxuan Du, Guoshun Yang, and Yanshuai Xu

In this manuscript, berry skins of 'Italia,' 'Benitaka,' 'Muscat of Alexandria,' 'Flame Muscat,' 'Rosario Bianco,' 'Rosario Rosso,' and 'Red Rosario' were used as research material throughout the stage of veraison (10 weeks after flowering and 11 weeks after flowering). RNA-Seq technology was used to analyse the relative expression of genes associated with the skin colour of grape berries.

Based on the results reported in this manuscript, it was concluded that MYBA1/2 and MYBPA1, key genes involved in the synthesis of grape anthocyanin, were highly expressed in red grape varieties, and the expression level was significantly lower in white grapes than in red grapes; The expression of the UFGT and LDOX genes was positively correlated with the MYBA key gene related to skin colour A new gene has been discovered (gene id: Vitvi19g01871) in this manuscript, which may play a key regulatory role in grape skin coloration. Further studies are proposed.

In the context of climate change, the role of such studies is becoming increasingly important.

The experiments were well-designed and carried out. The structure of the manuscript is basically sound, with only a few minor changes suggested (see below).

Detailed suggestions:

It is proposed to merge chapters 2.4 and 2.5 and to keep the title of chapter 2.5. „Statistical analyses”.

In line 33: „some fruit trees gradually develop inbred incompatibility;” Are you referring to sterility alleles? The situation is a bit different for grapes, that's what should be described.

Line 45-50: This section should be supplemented with examples of vine breeding. Here are some suggested references:

·         Sujata Tetali, S.P. Karkamkar and S.V. Phalake (2020): Mutation breeding for inducing seedlessness in grape variety ARI 516. International Journal of Minor Fruits, Medicinal and Aromatic Plants. Vol. 6 (2) : 67-71

·         Royo C, Torres-Pérez R, Mauri N, Diestro N, Cabezas JA, Marchal C, Lacombe T, Ibáñez J, Tornel M, Carreño J, Martínez-Zapater JM, Carbonell-Bejerano P. The Major Origin of Seedless Grapes Is Associated with a Missense Mutation in the MADS-Box Gene VviAGL11. Plant Physiol. 2018 Jul;177(3):1234-1253. doi: 10.1104/pp.18.00259. Epub 2018 May 31. PMID: 29853599; PMCID: PMC6053000.

line 54-56: You cite 7 references in one sentence; I suggest you explain them in more detail.

Line 77-79: Formatting error, please correct.

There are some typing errors, correct them carefully, please.

Throughout the manuscript, terms used in fruit growing are used, I suggest correcting this to terms used in viticulture (eg. use stock, training system etc.  instead of tree, tree form respectively)

In MDPI Horticulturae British English is used, I suggest using this, instead of US English.

Author Response

(The authors gave the same response as above.)

Round 2

Reviewer 1 Report

 As I mentioned before, grapevines are not TREES!! Please don’t make that mistake, if you like you could use the term fruit plants, but never fruit trees to described them.

I'm not familiar with Majorbio cloud platform, but you must include which tool was used to make the differential expression analysis (DEGs) or they must provided to you. Think in the idea that a publication must be like a cooking recipe, if you don't have one of the steps, you never will obtain the same results.

When you analyze a particular gene and you say that it is express differentially using the TPM values...I have to cite an explanation of TPM (https://hemtools.readthedocs.io/en/latest/content/NGS_pipelines/diff_genes.html):

"Basically, TPM is a technology-independent measurement because it is just a relative abundance, so it can be used to compare gene expression across different samples. However, in order to say a gene is truely differentially expressed, you have to have absolute gene expression, therefore, DESEQ2, EdgeR, sleuth, etc. need to be used for that purposes, they can give you a normalized TPM.

That means:

-to get differentially expressed genes/transcripts, we need to apply statistical tests, e.g. using sleuth

-for data visualization, e.g. heatmap, PCA, we can just use TPM and gene-level TPM (ref: Differential analyses for RNA-seq: transcript-level estimates improve gene-level inferences)"

Considering this, what you see in figures 5-8 could change when you do the correct analysis, also the result and conclusions.

Round 3

Reviewer 1 Report

Thanks for the opportunity of correct your work. I consider that this time is proper to pass.

From this exchange you are not the only ones who learned.

Nice work!